# Spreading of Isolated *Ptch* Mutant Basal Cell Carcinoma Precursors Is Physiologically Suppressed and Counteracts Tumor Formation in Mice

**DOI:** 10.3390/ijms21239295

**Published:** 2020-12-05

**Authors:** Nadine Brandes, Slavica Hristomanova Mitkovska, Dominik Simon Botermann, Wiebke Maurer, Anna Müllen, Hanna Scheile, Sebastian Zabel, Anke Frommhold, Ina Heß, Heidi Hahn, Anja Uhmann

**Affiliations:** Tumor Genetics Group, Institute of Human Genetics, University Medical Center Göttingen, Heinrich-Düker-Weg 12, 37079 Göttingen, Germany; nadine.brandes@med.uni-goettingen.de (N.B.); s.hristomanovamitk@stud.uni-goettingen.de (S.H.M.); dominik.botermann@med.uni-goettingen.de (D.S.B.); wiebke.maurer@stud.uni-goettingen.de (W.M.); anna.muellen@stud.uni-goettingen.de (A.M.); hannarabe@gmx.de (H.S.); sebastian.zabel@live.de (S.Z.); anke.frommhold@med.uni-goettingen.de (A.F.); ihess@gwdg.de (I.H.); hhahn@gwdg.de (H.H.)

**Keywords:** Patched receptor, basal cell carcinoma, keratin 5, epidermal cells, epidermis, hair follicle

## Abstract

Basal cell carcinoma (BCC) originate from Hedgehog/Patched signaling-activated epidermal stem cells. However, the chemically induced tumorigenesis of mice with a *CD4Cre*-mediated biallelic loss of the Hedgehog signaling repressor Patched also induces BCC formation. Here, we identified the cellular origin of *CD4Cre*-targeted BCC progenitors as rare Keratin 5^+^ epidermal cells and show that wildtype *Patched* offspring of these cells spread over the hair follicle/skin complex with increasing mouse age. Intriguingly, *Patched* mutant counterparts are undetectable in age-matched untreated skin but are getting traceable upon applying the chemical tumorigenesis protocol. Together, our data show that biallelic *Patched* depletion in rare Keratin 5^+^ epidermal cells is not sufficient to drive BCC development, because the spread of these cells is physiologically suppressed. However, bypassing the repression of *Patched* mutant cells, e.g., by exogenous stimuli, leads to an accumulation of BCC precursor cells and, finally, to tumor development.

## 1. Introduction

Epidermal stem cells of the hair follicle (HF) outer root sheath (ORS) [1], bulge [2], secondary hair germ [3,4,5] and/or the interfollicular epidermis (IFE) [1,6] can give rise to basal cell carcinoma (BCC). BCC are the most frequent skin neoplasia in humans [7], with mutations in the Hedgehog (Hh)-signaling inhibitor Patched (Ptch) appearing as one of the major driving forces in the development of this tumor entity [8]. We recently showed that a *CD4Cre*-mediated homozygous *Patched* deletion (*Ptch^f/f^ CD4Cre*) does not affect T-cell functions in vitro and in vivo [9,10] but can result in BCC formation upon treatment with 7,12-Dimethylbenz[a]anthracene (DMBA)/12-O-tetradecanoylphorbol-13-acetate (TPA) [11]. This was an unexpected observation, because *CD4Cre*-deleter mice express a *Cre*-recombinase gene under the control of the *Cluster of differentiation 4* promoter/enhancer/silencer. However, *Ptch^f/f^ CD4Cre* mice do not spontaneously develop BCC [11], demonstrating that homozygous *Ptch* depletion using the *CD4Cre* driver is not sufficient for BCC development. This is in contrast to BCC mouse models using “classical” BCC drivers [4,12,13], in which the *Ptch* mutation is simultaneously induced in a large proportion of HF stem cells [4] or basal IFE cells [1,4,13] and in which BCC develop spontaneously. These data suggest that the *CD4Cre* driver targets BCC progenitors with a lower tumorigenic potential and/or at lower frequency compared to “classical” BCC models. However, this also opens the question whether a certain quantity of Hh-activated BCC precursors is necessary for BCC development, which could have far-reaching consequences for the understanding and treatment of sporadic human BCC.

We here determined the *CD4Cre*-targeted BCC progenitor cell type and its frequency by following the nonhematopoietic progeny of wildtype *Ptch* and *Ptch* mutant *CD4Cre*-targeted cells in the adult skin by lineage tracing experiments. We show that the *CD4Cre* transgene is expressed in K5^+^ epidermal cells and that wildtype *Ptch* progenies of *CD4Cre*-targeted keratinocytes populate the adult HF/skin complex with increasing mouse age. In contrast, *Ptch* mutant progenies of *CD4Cre*-targeted keratinocytes are undetectable and do not accumulate like their wildtype *Ptch* counterparts under normal conditions. However, the exogenous stimulation of their survival can result in BCC development. Taking together, our data demonstrate that isolated BCC precursors with a homozygous *Ptch* depletion do not spread or accumulate and are not sufficient for BCC development under normal conditions.

## 2. Results

### 2.1. Wildtype Ptch Progeny of CD4Cre-Targeted Cells Spread over the HF/Skin Complex with Increasing Mouse Age

To characterize the cellular origin of *Ptch^f/f^ CD4Cre* BCC, *CD4Cre*-targeted wildtype *Ptch* cells were traced in *CD4Cre R26-tdT* mice by their tdTomato (tdT) expression (see Appindix A, Figure A1 for tdT expression in *CD4Cre R26-tdT* thymus). Flow cytometric analyses revealed the existence of various tdT-expressing cell populations in back skin epidermal isolates of *CD4Cre R26-tdT* mice in comparison to the controls (Figure 1A,B). Based on the expression levels of tdT and the general keratinocyte marker CD49f, we observed three relatively stable tdT-expressing populations (tdT^+^ CD49f^low^, tdT^+^ CD49f^high^ and tdT^low^ CD49f^high^) and a tdT^high^ CD49f^high^ population, which strongly augments with increasing mouse age (Figure 1A,C). Further analyses revealed that the tdT^+^CD49f^low^ population mainly consists of TCRβ-, CD3- or CD16-expressing immune cells (Figure 2), whereas the tdT^+^ CD49f^high^ and the tdT^low^ CD49f^high^ populations contain small numbers of CD16-expressing (e.g., macrophages; Figure 2) or TCRβ and CD3-expressing immune cells (T cells; Figure 2), respectively (for the verification of antibody specificity, see Appendix A
Figure A2). Remarkably, tdT^high^ CD49f^high^ cells do not express immune cell markers (Figure 2), indicating that this population has a pure keratinocyte identity. To verify our conclusion that the number of keratinocytes descending from *CD4Cre*-targeted cells increases with mouse age, we analyzed whole mount preparations from the back skin of differentially aged *CD4Cre R26-tdT* mice. Indeed, this approach showed that the skin of aged *CD4Cre R26-tdT* mice contains enormous numbers of wildtype *Ptch* tdT^+^ HF compared to younger mice (Figure 3A), whereas, in the third anagen of *CD4Cre R26-tdT* back skin (11 weeks old), only isolated tdT^+^ HF were detected; the numbers of tdT^+^ HF increased enormously from the fourth (16 weeks old), fifth (25 weeks old) and to the ninth anagen (55 weeks old) (Figure 3A). Thereby, tdT^+^ cells grow over the entire length of anagen HF (Figure 3B) and, also, in the IFE of *CD4Cre R26-tdT* back skin (Figure 3C), indicating that the *CD4Cre* transgene targets cells of the HF and of the IFE compartment.

### 2.2. The CD4Cre-Deleter Targets Keratin 5^+^ Epidermal Cells That Are the Origin of DMBA/TPA-Induced BCC in Ptch^f/f^ CD4Cre Mice

Both the ORS of HF and the basal layer of the IFE are characterized by Keratin 5 (K5) expression [1]. Thus, lineage tracing of the *K5* promoter-driven CreERT-recombinase expression results in the labeling of cells in both compartments [14]. Furthermore, since K5^+^ cells can be the origin of BCC [1] (for review, see [15]), we hypothesized that the *CD4Cre* transgene targets K5^+^ epidermal cells. Indeed, immunofluorescence stainings of individually isolated tdT^+^ HF and cryo-sectioned back skin of *CD4Cre R26-tdT* mice revealed that K5^+^ cells of the ORS (Figure 4A), as well as of the basal layer, express tdT (Figure 4B). Moreover, in vitro cultured tdT^neg^ keratinocytes from *CD4Cre R26-tdT* back skin started to express tdT 22 days post-seeding. The subsequent immunofluorescent staining verified that all newly recombined tdT^+^ cells express K5 (Figure 4C). Forty-two or 55 days after seeding, 0.47% or 0.68% of all K5^+^ cells were tdT^+^, respectively. Based on these data, we concluded that the *CD4Cre* driver targets rare K5^+^ keratinocytes, which most probably are also the origin of BCC in *Ptch^f/f^ CD4Cre* mice. If this is the case, the histological appearance of the DMBA/TPA-induced BCC in *Ptch^f/f^ CD4Cre* mice should mimic that of BCC from *Ptch^f/f^ K5CreERT* mice. The latter arise from the lower HF and from the IFE [13,15] and express K5 (Appendix A
Figure A3). However, BCC of DMBA/TPA-treated *Ptch^f/f^ CD4Cre* skin (for experimental set-up, see Figure 5A) exclusively occur at the IFE in HF-near regions (Figure 5B,C) and express K5 (Figure 5C) but never grow as tumors of the bulge or secondary hair germ like in *Ptch^f/f^ K5CreERT* skin [13] (Appendix A
Figure A3). To evaluate the possibility that the chemical treatment may preferably induce BCC development from IFE cells, we furthermore analyzed BCC from DMBA/TPA-treated heterozygous *Ptch^+/−^* mice [16]. Contrarily to BCC in *Ptch^f/f^ CD4Cre* skin, and similar to BCC from *Ptch^f/f^ K5CreERT*, BCC from DMBA/TPA-treated *Ptch^+/−^* mice arise from HF and IFE (Figure 5D,E; for a comparison of BCC/mm skin in DMBA/TPA-treated *Ptch^f/f^ CD4Cre* and *Ptch^+/−^* mice, see Appendix A
Figure A4) and stain positive for K5 (Figure 5E). This shows that the origin of DMBA/TPA-induced BCC is not determined by the chemical treatment but, rather, by the compartment of the genetically targeted cell type, and thus, the *CD4Cre*-deleter most likely targets K5^+^ basal cells of HF-near IFE.

### 2.3. Isolated Ptch Mutant Epidermal Cells do Not Spread Like Their Wildtype Ptch Counterparts

The *CD4Cre*-deleter targets rare K5-expressing epidermal IFE cells, in which a biallelic *Ptch* mutation did not, per se, result in BCC formation. This indicates that the quantity of K5^+^
*Ptch* mutant BCC precursors in untreated *Ptch^f/f^ CD4Cre* skin is not sufficient for spontaneous BCC development, as seen in *Ptch^f/f^ K5CreERT* mice. However, our lineage-tracing analyses showed that robust numbers of wildtype *Ptch* epidermal keratinocytes descending from *CD4Cre*-trageted cells grow in the skin of aged mice. Thus, one could speculate that the numbers of *Ptch* mutant keratinocytes growing in the skin of aged *Ptch^f/f^ CD4Cre* mice are comparable, which would oppose the assumption that a low quantity of K5^+^
*Ptch* mutant BCC precursor is not sufficient for BCC development. To shed light on this discrepancy, we traced *Ptch* mutant progenies of *CD4Cre*-targeted cells in *R26-LacZ Ptch^f/f^ CD4Cre* skin. Remarkably, and in contrast to wildtype *Ptch* skin (Figure 6A), this approach revealed that HF and IFE of 42-week-old *R26-LacZ Ptch^f/f^ CD4Cre* mice (Figure 6B), as well as of *R26-LacZ Ptch^f/f^ CD4Cre* mice transplanted with wildtype *Ptch* bone marrow (Figure 6C), are completely free of progenies of *CD4Cre*-targeted cells (control LacZ stainings, Appendix A
Figure A5). In contrast, DMBA/TPA-treated *R26-LacZ Ptch^f/f^ CD4Cre* skin shows *Ptch* mutant LacZ^+^ HF (Figure 6B) but at much lower numbers compared to the skin of age-matched untreated *CD4Cre R26-tdT* mice, which show numerous wildtype *Ptch* tdT^+^ HF (Figure 6A). Anti-K5 antibody staining furthermore revealed that the few LacZ^+^ cells in HF and HF-near IFE of DMBA/TPA-treated *R26-LacZ Ptch^f/f^ CD4Cre* skin are positive for K5 or at least descend from K5^+^ cells (e.g., suprabasal layers of the IFE) (Figure 6D).

These findings indicate that isolated *Ptch* mutant descendants from K5^+^ epidermal cells do not contribute to normal skin homoeostasis, since the spreading of these rare epidermal cells (e.g., of *CD4Cre*-targeted cells) is suppressed under physiological conditions. Thus, as a result, the probability of spontaneous BCC development is strongly reduced in *Ptch^f/f^ CD4Cre* skin. In addition, whereas the *Ptch* status of T cells does not influence this phenomenon, the DMBA/TPA treatment seems to enhance the survival probability of *Ptch* mutant K5^+^ epidermal cells and their offspring, which, finally, can result in BCC formation.

## 3. Discussion

The development of BCC is closely related to *Ptch* mutations and the subsequent activation of Hh/Ptch signaling in stem cells of the HF and the IFE [8]. However, genome-sequencing data from human BCC samples hint towards a more complex pathway interaction in BCC development [17]. In line with this assumption, our novel data show that homozygous *Ptch* depletion in rare K5^+^ epidermal cells is insufficient to drive BCC formation in mice, most likely due to the physiological erasure of the mutant cells.

In the skin, as an organ prone for Hh/Ptch-associated BCC development, a biallelic *Ptch* mutation, which occurs simultaneously in all stem cells of the HF or of the IFE, leads to BCC formation [4,13]. Thus, the lack of a spontaneous BCC development in *Ptch^f/f^ CD4Cre* mice, together with our novel results that wildtype *Ptch CD4Cre*-targeted epidermal cells widely populate the HF/skin complex, were puzzling. However, our data also revealed that neither untreated nor DMBA/TPA-treated *Ptch^f/f^ CD4Cre* skin contains comparable amounts of *CD4Cre*-targeted offspring, as seen in aged wildtype *Ptch* skin. Thus, under normal physiological conditions, BCC development in *Ptch^f/f^ CD4Cre* skin seems to be repressed by the physiological inhibition of the spreading of rare *Ptch* mutant epidermal cells. Since *CD4Cre*-mediated *Ptch* depletion does not affect T cell (e.g., cytotoxic T cells [CTL]) function in vivo [9,10], and wildtype *Ptch* bone marrow transplantation does not induce a spread of *Ptch* mutant keratinocytes in *Ptch^f/f^ CD4Cre* mice [11] (see Figure 6C), an immune cell-mediated suppression of *Ptch* mutant keratinocytes can be excluded. However, hypothetically, an epithelial defense against cancer (EDAC), which is based on cell competition between cells of different fitness [18], might mediate the inhibition of *Ptch* mutant keratinocyte spreading. One major prerequisite for the elimination of less fit cells (or mutant cells) by EDAC is that the genetically altered ones are surrounded by wildtype cells [18]. This is most likely the case for *Ptch* mutant epidermal cells in *Ptch^f/f^ CD4Cre* skin. However, various parameters (e.g., ratio of normal/transformed cells and environmental factors) can affect the EDAC and potentially convert the antitumorigenic process to a super-competition, in which cells with an accumulation of a series of oncogenic mutations outcompete normal or single-mutant cells [18]. Indeed, we observed that the DMBA/TPA treatment of *R26-LacZ Ptch^f/f^ CD4Cre* skin increases the probability to detect LacZ^+^
*Ptch* mutant epidermal cell clusters in isolated HF and IFE areas that have not yet developed to BCC. These observations indicate that the chemicals confer a survival benefit of the rare *Ptch* mutant K5^+^ epidermal cells in *Ptch^f/f^ CD4Cre* skin, which might be due to an accumulation of oncogenic mutations and the suppression of normal cells. Probably, aberrant Ras signaling does not play a role in DMBA/TPA-induced BCC development in *Ptch^f/f^ CD4Cre* mice [11], while the disruption of physiological apoptotic responses via the DMBA/TPA-mediated downregulation of p53 [19] is more likely [12,20,21]. Moreover, the induction of apoptosis in BCC cells, e.g., by Fas upregulation, reduces the development and the size of UV light-induced BCC [20]. Hence, in the case of a homozygous *Ptch* mutation in isolated BCC precursors, the DMBA/TPA treatment, similarly to UV light exposure [15,22,23], potentially may abrogate apoptotic processes, which normally can erase isolated *Ptch* mutant cells from the skin. Subsequently, *Ptch* mutant epidermal cells are multiplied, accumulate additional mutations and, thus, are predisposed to BCC development. Indeed, *Ptch* heterozygous *SKH1 Hairless* mice progressively develop spontaneous BCC, which are further boosted by UV light exposure. This suggests that, irrespective of aberrant Hh/Ptch signaling, age-related effects (e.g., altered DNA damage responses) play a crucial role in BCC formation [22,23].

The fact that a homozygous *Ptch* mutation in isolated K5^+^ epidermal cells is not sufficient for BCC development also questions the current BCC therapy based on Hh signaling inhibitors (e.g., vismodegib). Whereas these inhibitors only target Hh/Ptch signaling and suppress the growth of the initial BCC precursor cell, they will not target secondly accumulated tumor-promoting cascades. Indeed, cessation of the vismodegib treatment of *Ptch*/*p53*-mutant BCC not only leads to a loss of Hh-signaling marker gene expression but, also, to changes of HF stem cell-like to IFE- and isthmus-like expression profiles of the remaining BCC [21]. In humans, these inhibitors can lead to the development of BCC-adjacent cSCC (cutaneous squamous cell carcinoma) [24,25,26], which show decreased Hh/Ptch but increased Ras/MAPK signaling [27]. Moreover, the genetic differences of pretreatment BCC and post-treatment cSCC are minor (3%) [27], and mutations associated with cSCC development (e.g., in effectors of the Hippo-YAP pathway and in MYCN) [28,29,30,31] have also been identified in BCC [17]. This strongly argues for a scenario in which a complex pathway interaction drives the BCC development. Such a scenario is also supported by our data demonstrating that the spread of isolated biallelic *Ptch* mutant BCC precursors is suppressed under normal conditions (potentially by an EDAC) and that they only develop into BCC upon a second nonphysiological stimulus. Moreover, in light of the recently made postulation that, in human skin, spatial relationships and competitive behaviors amongst clones act as suppressors of malignant progression [32], the *Ptch^f/f^ CD4Cre* BCC mouse model is more closely related to the human situation than “classical” BCC mouse models. This is due to the fact that the *CD4Cre*-deleter targets only single K5^+^ epidermal cells, whereas “classical” skin-specific drivers target all or large proportions of HF stem cells and/or basal cells simultaneously [4,13]. Thus, new BCC models such as *Ptch^f/f^ CD4Cre* mice will not only allow to investigate BCC-initiating events beyond Hh/Ptch-signaling activation but, also, to develop new BCC treatment options.

## 4. Materials and Methods

### 4.1. Mice

All animal experiments were performed in compliance with German legal and ethical requirements and were approved by the Lower Saxony State Office for Consumer Protection and Food Safety (file numbers 33.9-42502-04-15/1926 from November 2015, 33.9-42502-04-11/0374 from April 2011 and 33.14-42502-04-100/07 from January 2008). The following mouse strains were used: *Tg(Cd4-cre)1Cwi/Bflu* (*CD4Cre*, JAX stock #017336) [33,34,35], *K5CreERT* [36], *Gt(ROSA)26Sor^tm9(CAG-tdTomato)Hze^* (*R26-tdT,* JAX stock #007905) [37], *Gt(ROSA)26Sor* (*R26-LacZ*, JAX stock #003309) [38], *Gt(ROSA)26Sor^tm1(cre/ERT2)Tyj^* (*R26-CreERT2*, JAX stock #008463) [39], *TgN(beta-act-EGFP)* [40], *Ptch1^tm1Zim^* (*Ptch^+/−^*) [41] and *Ptch1^tm1Hahn^* (*Ptch^f/f^*, JAX stock #012457) [42]. All used mouse strains were maintained on a C57BL/6 background. Genotyping was conducted by PCR on genomic DNA isolated from tail biopsies using primer pairs recommended by the providing scientists or by The Jackson Laboratory. Both genders of transgenic mice were used. No sex-specific differences were observed.

*R26-LacZ* and *R26-CreERT2* mice were intraperitoneally (i.p.) injected with 1-mg tamoxifen in sunflower oil for 5 consecutive days at an age of 8 weeks [42]. DMBA/TPA treatment (Sigma-Aldrich Inc., St. Louis, MO, USA) and adoptive bone marrow transfer to irradiated mice were described previously [11].

### 4.2. Isolation of Keratinocytes, Epidermal Sheets and Individual Hair Follicles

Murine epidermal cells for flow cytometric analyses and in vitro culture were isolated using thermolysin as previously described [11]. Epidermal sheets of murine back skin and individual HF were isolated according to [43,44], respectively.

### 4.3. Tissue Embedding and Sectioning

Tissue samples were fixed in 4% paraformaldehyde/1x phosphate buffered saline (PBS) at 4 °C for 2 or 3 days and, depending on the subsequent analyses, either dehydrated and embedded in paraffin or equilibrated overnight in 20% sucrose/1x PBS at 4 °C and embedded in cryo-medium (Medite, Burgdorf, Germany). Paraffin-embedded or cryo-embedded samples were sectioned using a microtome or a cryotome, respectively, and used for antibody or H&E staining.

### 4.4. Primary Keratinocyte Culture

For primary keratinocyte culture, epidermal cells were isolated from telogen back skin of 19-week-old mice and were cultured under feeder-free conditions on collagen-coated dishes (Bovine Collagen Solution Type I, Sigma Aldrich Inc., St. Louis, MO, USA) in defined keratinocyte-serum-free medium (DK-SFM) basal medium (supplemented with DK-SFM growth supplements, Thermo Fisher Scientific, Waltham, MA, USA) for up to 55 days without passaging.

### 4.5. LacZ Staining, Antibody Staining, Flowcytometric Analyses and Microscopy

LacZ(β-galactosidase-) staining of tissue samples, immunohistological and immunofluorescent antibody stainings of paraffin and cryotome sections and fixed in vitro cultured cells, as well as flow cytometric analyses of keratinocytes, were described previously [11,45,46]. For antibody staining of individual HF, the samples were blocked for 30 min in 0.2% I-Block (Applied Biosystems, Foster City, CA, USA) and stained overnight at 4 °C with primary antibodies in 1 × tris buffered saline (TBS)/0.1% triton X-100, followed by a 4-h washing step in 1 × TBS/0.1% triton X-100. Samples were incubated overnight at 4 °C with secondary antibodies in 1 × TBS/0.1% triton X-100 and washed 4× for 1 h in 1 × TBS/0.1% triton X-100.

Fluorescent or immunohistological stainings or whole-organ analyses were documented on a confocal laser scanning microscope equipped with the software FluoView FV100 (Olympus Corporation, Shinjuku, Japan), Olympus BX60 microscope equipped with cellSense software or a fluorescent dissecting microscope (Leica M205FA) equipped with a digital camera (Leica DFC 450C) and the software Leica Application Suite, respectively. Flow cytometric analyses were conducted on a LSR II flow cytometer (BD Biosciences Pharmingen, San Jose, CA, USA). Data acquisition and analyses were performed using BD FacsDiva (BD Biosciences Pharmingen, USA) and FlowJo (Treestar, Ashland, OR, USA) software. If not otherwise stated, for each analysis, 2.5 × 10^6^ keratinocytes were counted. For quantification of tdT^low^ CD49f^high^, tdT^+^ CD49f^high^ and tdT^high^ CD49f^high^ cell numbers, anti-CD49f-PerCP-Cy5.5 antibody-stained back skin isolates of differentially aged *CD4Cre R26-tdT* mice (N_2nd hair cycle_ = 5, N_4th hair cycle_ = 3 and N_9th hair cycle_ = 3) were analyzed by flow cytometry. Used antibodies, antibody concentrations and retrieval methods are summarized in Appendix A
Table A1 and Table A2.

## 5. Conclusions

The spread of rare *Ptch* mutant K5^+^ epidermal stem cells is physiologically suppressed. These cells only progress to BCC upon events that counteracted this suppression. These findings strengthen the more recent assumption that BCC rather develop due to a complex pathway interaction than to sole Hh/Ptch-signaling overactivation. Additionally, it indicates that mouse models for targeting isolated stem cells of the skin are needed to more precisely reflect sporadic human BCC development, which will help to better understand BCC-initiating events and to develop new, targeted BCC treatment options.

## Figures and Tables

**Figure 1 ijms-21-09295-f001:**
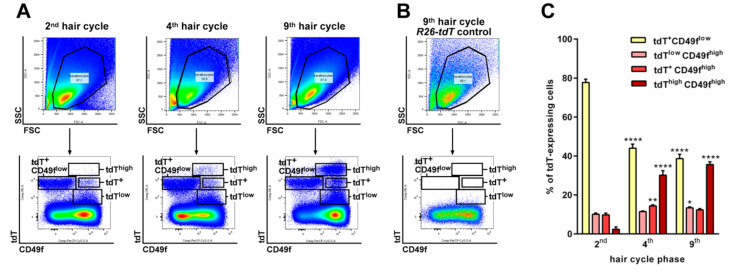
tdTomato (tdT)^high^ CD49f^high^-expressing cells accumulate in *CD4Cre R26-tdT* skin with increasing mouse age. (**A**,**B**) Representative flow cytometric analyses of (**A**) 2,500,000 *CD4Cre R26-tdT* back skin isolates of the 2nd, 4th and 9th telogens and (**B**) 1,000,000 *R26-tdT* control back skin isolates of the 9th telogen stained with anti-CD49f-peridinin-chlorophyll-protein (PerCP)-Cy5.5 antibodies. Top: Forward scatter (FSC)/side scatter (SSC) plots for gating on living cells. Bottom: tdT (phycoerythrin [PE] channel)/CD49f plots for visualization of the tdT expression on the gated living cells. In *CD4Cre R26-tdT* back skin isolates, 4 different tdT-expressing populations (one CD49f^low^-expressing population: tdT^+^ CD49f^low^ and 3 CD49f^high^-expressing populations: tdT^low^, tdT^+^ and tdT^high^) were distinguishable in differential aged mice, whereas no tdT^+^ cells were detected in *R26-tdT* back skin isolates using the PE channel (**B**). (**C**) Percentage share of tdT^+^ CD49f^low^ and tdT^low^, tdT^+^ and tdT^high^ cells (N_2nd_ = 5, N_4th_ = 3 and N_9th_ = 3) at the indicated hair cycle phases (based on the flow cytometric analyses shown in (**A**)). Bars represent mean +/− SEM. Significant differences were calculated using a nonparametric Mann-Whitney test. * *p* > 0.05, ** *p* > 0.01 and **** *p* > 0.0001.

**Figure 2 ijms-21-09295-f002:**
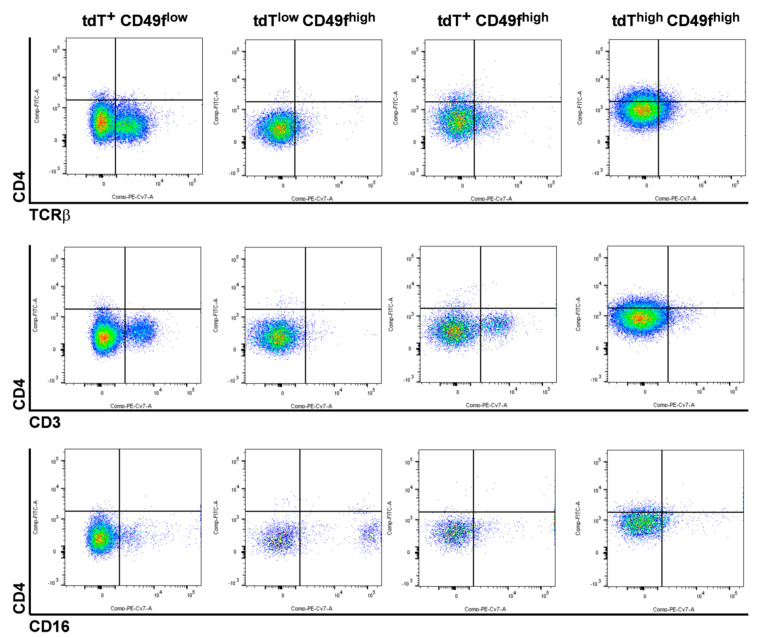
tdT^high^ CD49f^high^-expressing cells of aged *CD4Cre R26-tdT* mice do not express immune cell markers. Representative plots of flow cytometric analyses of 2,500,000 *CD4Cre R26-tdT* back skin isolates of the 4th telogen stained with anti-CD4-fluorescein isothiocyanate (FITC); anti-CD49f-PerCP-Cy5.5 and anti-TCRβ-PE-Cy7 (top), CD3-PE-Cy7 (middle) or anti-CD16-PE-Cy7 (bottom) antibodies gated as shown in Figure 1A. Individual CD4/TCRβ, CD4/CD3 or CD34/CD16 analyses of the tdT^+^ CD49f^low^ and the tdT^low^ CD49f^high^, tdT^+^ CD49f^high^ and tdT^high^ CD49f^high^ subpopulations revealed TCRβ^+^ and CD3^+^ immune cells in the tdT^+^ CD49f^low^ and the tdT^+^ CD49f^high^ populations, whereas CD16^+^ immune cells were detected in the tdT^+^ CD49f^low^ and the tdT^low^ CD49f^high^ populations. tdT^high^ CD49f^high^ cells express none of the immune cell markers, except for a slightly increased CD4 protein level.

**Figure 3 ijms-21-09295-f003:**
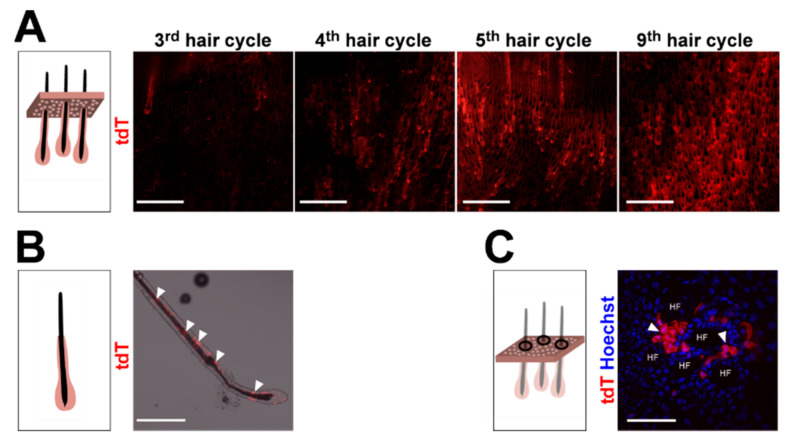
Wildtype *Ptch* tdT^+^ cells accumulate in hair follicle (HF) and interfollicular epidermis (IFE) with increasing age of the *CD4Cre R26-tdT* mice. (**A**–**C**) Representative fluorescent analyses of (**A**) epidermal sheets of the back skin of *CD4Cre R26-tdT* mice at the indicated hair cycles (dermal view), (**B**) an individually isolated HF of the back skin of a *CD4Cre R26-tdT* mouse and (**C**) an epidermal preparation at late-anagen of the back skin of a *CD4Cre R26-tdT* mouse (top view). Scale bars: 1 mm (**A**) and 100 µm (**B**,**C**).

**Figure 4 ijms-21-09295-f004:**
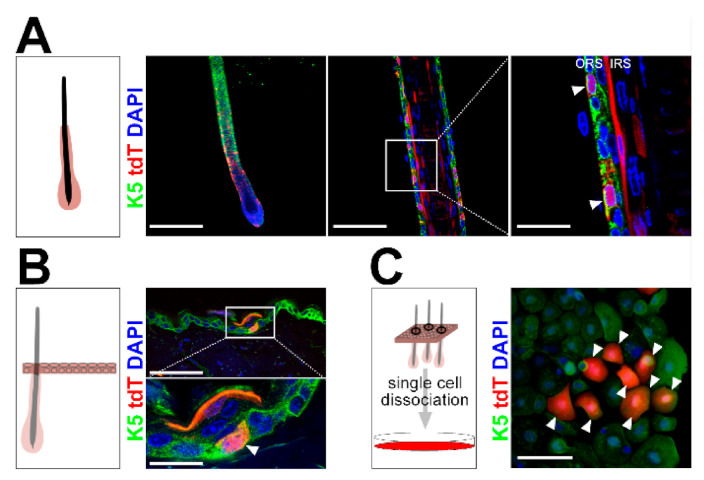
Wildtype *Ptch* tdT^+^ cells of *CD4Cre R26-tdT* back skin descents from *CD4Cre*-targeted rare K5^+^ epidermal cells. (**A**,**B**) Representative fluorescent analyses of (**A**) an individually isolated HF of back skin and (**B**) cryo-sectioned back skin of *CD4Cre R26-tdT* mice stained against K5. (**C**) In vitro cultured tdT^neg^ epidermal cells of *CD4Cre R26-tdT* back skin 55 days post-seeding stained against the outer root sheath (ORS) and basal cell marker K5. tdT^+^ K5^+^ double-positive cells are marked with arrow heads. Box: zoomed area. IRS: inner root sheath. Scale bars: 100 µm (**A**, left and **C**), 33 µm (**A**, middle and **B**, top) and 10 µm (**A**, right).

**Figure 5 ijms-21-09295-f005:**
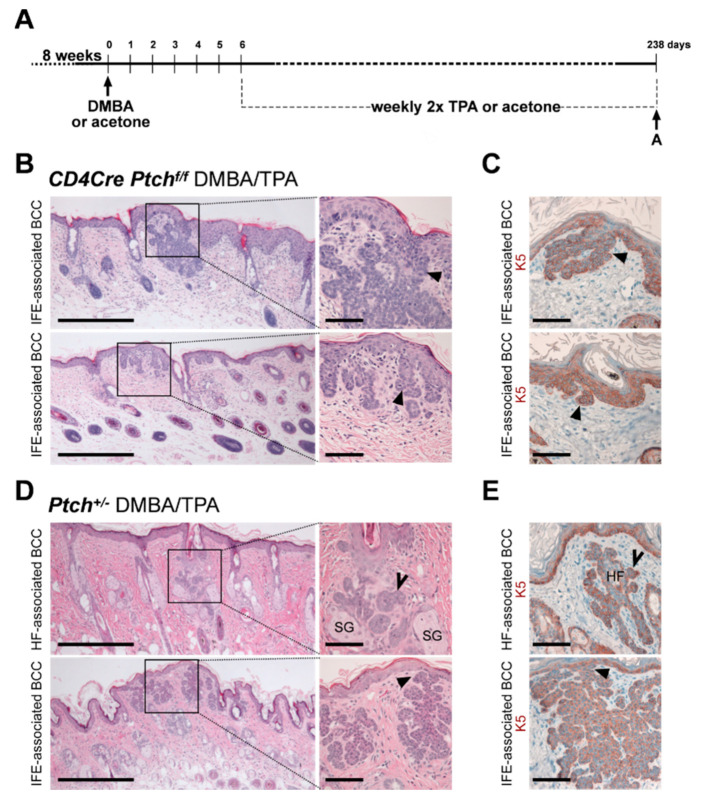
Basal cell carcinoma (BCC)-like tumors of 7,12-Dimethylbenz[a]anthracene (DMBA)/12-O-tetradecanoylphorbol-13-acetate (TPA)-treated *Ptch^f/f^ CD4Cre* skin appear exclusively IFE-associated. (**A**) Experimental set-up of the DMBA/TPA carcinogenesis protocol. (**B**–**E**) Representative hematoxylin and eosin (H&E) stainings (**B**,**D**) and anti-K5 antibody stainings (**C**,**E**) of BCC from 42-week-old DMBA/TPA-treated *Ptch^f/f^ CD4Cre* (**B**,**C**) and *Ptch^+/−^* back skin (**D**,**E**) (N*_Ptchf/f CD4Cre_* = 13 and N*_Ptch+/−_* = 8). Solid arrows: BCC associated to the IFE; open arrows: BCC associated to HF. HF: hair follicle, SG: sebaceous gland. Boxes: zoomed areas. Scale bars: 20 µm (**B**, left and **D**, left) and 100 µm (**B**, right, **C** and **D**, right and **E**).

**Figure 6 ijms-21-09295-f006:**
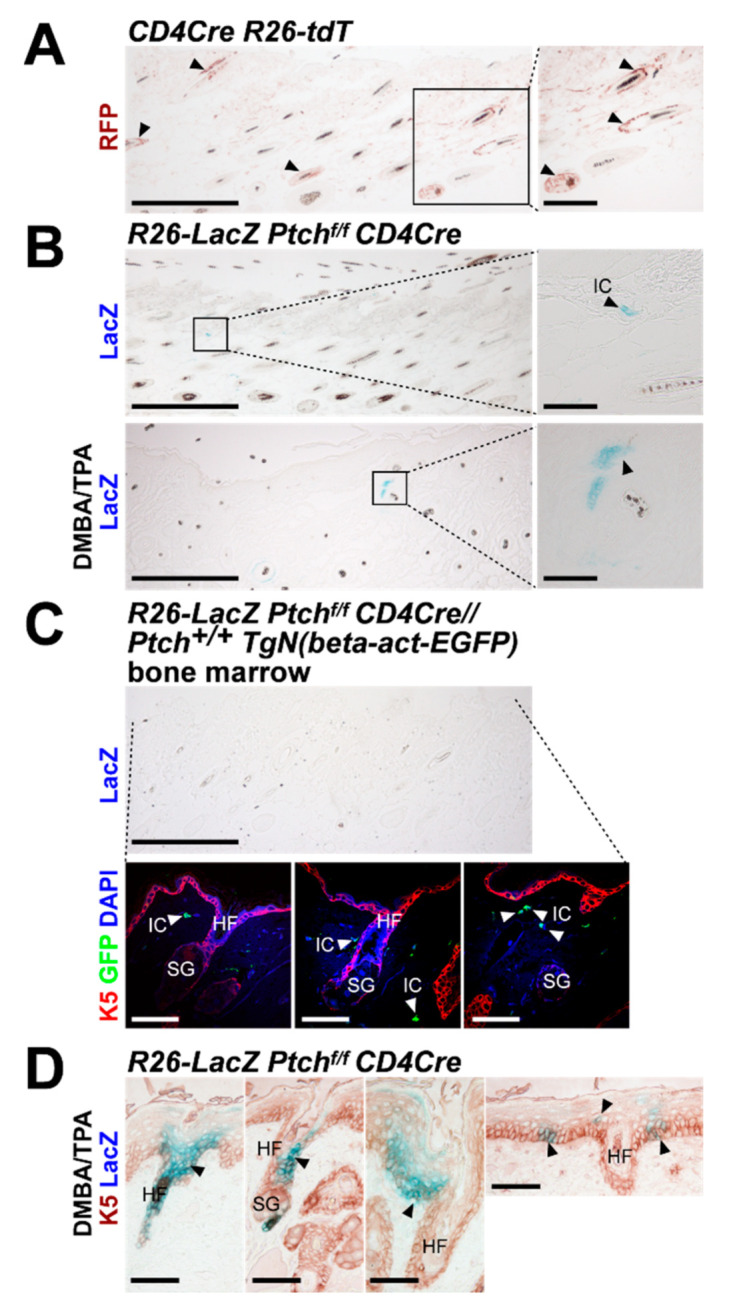
*Ptch* mutant descendants of *CD4Cre*-targeted epidermal cells do not populate the skin with increasing mouse age like their wildtype *Ptch* counterparts. (**A**,**B**) Representative anti-red fluorescent protein (RFP)-stained paraffin sections of (**A**) *CD4Cre R26-tdT* back skin and (**B**) LacZ-stained untreated (top) and DMBA/TPA-treated (bottom) *R26-LacZ Ptch^f/f^ CD4Cre* back skin at an age of 42 weeks (N*_CD4Cre R26-tdT_* = 7; N*_R26-LacZ_*
_*Ptchf/f CD4Cre*_ = 6). The tdT protein was visualized using an anti-RFP antibody. (**C**) Representative anti-K5/anti-green fluorescent protein (GFP) antibody/LacZ-stained paraffin sections of untreated back skin of a *R26-LacZ Ptch^f/f^ CD4Cre* mouse adoptively transplanted with wildtype *Ptch TgN(beta-act-EnhancedGFP)* bone marrow at an age of 42 weeks (30 weeks after transplantation). (**D**) Representative anti-K5 antibody/LacZ-stained paraffin sections of DMBA/TPA-treated *R26-LacZ Ptch^f/f^ CD4Cre* back skin at an age of 42 weeks. Black arrow heads: tdT^+^ LacZ^+^ or double-positive cells/areas and white arrow heads: GFP^+^ immune cells. Boxes: zoomed areas. IC: immune cells; HF: hair follicle; SG, sebaceous gland. Scale bars: 20 µm (**A**, left, **B**, left and **D**, top); 100 µm (**A**, right, **B**, right and **C**) and 33 µm (**D**, bottom).

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
