# Peer review of "Spreading of Isolated *Ptch* Mutant Basal Cell Carcinoma Precursors Is Physiologically Suppressed and Counteracts Tumor Formation in Mice"

_ijms, 2020, doi:10.3390/ijms21239295_

Round 1

Reviewer 1 Report

This study is a follow up of a previous report by the Hahn group that shows that CD4-Cre;Ptc1fl/fl mice develop BCC when the skin is treated with DMBA/TPA. Here, the authors show that, unlike control CD4-Cre; R26-tDt keratinocytes, the cells with loss of Ptc1 do not accumulate as the mice age unless tumourigenesis is chemically stimulated.

The major problem I have with the interpretation of the findings that the Ptc1 knockout cells do not spread while the control counterparts do, is that I believe the most logical explanation is that enhanced Hh signalling in CD4+ lymphocytes, including CD4+ cytotoxic lymphocytes (CTLs) could easily explain those findings. However, this likely scenario is not even discussed, while a different hypothesis of epithelial defines against cancer is proposed. The genetic model should delete Ptc1 in the CD4+ immune cells, activating Hedgehog signalling in CD4+ CTLs, known to increase their cytotoxic effects. Thus, an enhanced immune system is more likely to keep mutant Ptc1 keratinocytes in check, preventing cancer formation. Although the formation of DMBA/TPA BCCs in these mice is not affected by bone marrow transplant of wt mice (as shown in the previous publication in JID), the authors should determine here if a bone marrow transplant results in enhanced spreading of the mutant skin cells to a level comparable to the control, or not. This is a key experiment to understand the mechanism that controls the proliferation/survival of the rare keratinocyte population affected by the CD4-Cre recombination.

While the results are well presented, the study has reduced novelty, since most of the observations are confirmation of the results published by the group and others, and the most interesting and novel finding needs more work to be conclusive.

It is also essential that the authors describe in more detail their previous publication and that they revise the style to improve clarity, as several parts are hard to read.

Another minor but very important concern is the inaccurate nomenclature of the Patched1 gene in mice. It should be abbreviated Ptc1 everywhere, since there are two homologues, Ptc1 and Ptc2, and the nomenclature including the "h" is used for human PTCH1 and PTCH2.

Author Response

Answer to query 1: We apologize for not having mentioned the fact that we already have shown that the CD4Cre-mediated Ptch depletion did not impact the function of T cells (e.g. CTL, Treg) in vitro and in 3 different in vivo disease models (incl. adoptively transferred melanoma cells to test tumor surveillance by the immune system) [1, 2]. In contrast to de la Roche et al. who showed that Smo depletion (and thus Hh signaling inactivation) decreases the cytotoxic function of CTL in vitro [3], we neither detected increased cytotoxicity of Ptch-depleted CTL nor an altered tumor surveillance by the adaptive immune system in Ptchf/f CD4Cre mice compared to the controls [1]. Thus, our data clearly showed that Ptch is dispensable for T cell function [1, 2]. Beyond that, we now provide histological analysis of untreated skin of R26-LacZ Ptchf/f CD4Cre mice transplanted with wildtype Ptch bone marrow. These data show that the skin of these mice, comparable to Ptchf/f CD4Cre mice, completely lack LacZ+ Ptch mutant keratinocytes demonstrating that spreading of Ptch mutant keratinocytes is suppressed independently of the Ptch status (wildtype or mutant) of T cells. The new data are now included into the revised Figure 6 and are described in lines 125-126 in the Result section. We furthermore added a paragraph to the Introduction section (lines 41-42) describing our previous results that Ptch is dispensable for T cell function and discuss our novel results in light of our former data (lines 214-219). We furthermore reviewed the manuscript for spelling errors and highlighted all changes by grey mark ups.

Answer to query 2: We did not follow the suggestion of the reviewer since according to Mouse Genome Informatics (MGI; informatics.jax.org) the murine Patched1 gene is abbreviated Ptch1 or Ptch. Moreover, the international nomenclatures of the transgenic mouse models used in our study are Ptch1tm1Hahn and Ptch1tm1Zim. The abbreviation Ptc1 is dedicated for the Drosophila melanogaster Patched1 gene (see Flybase.org).

References

  1. Michel, K.D., et al., The hedgehog receptor patched1 in T cells is dispensable for adaptive immunity in mice. PLoS One, 2013. 8(4): p. e61034.
  2. Uhmann, A., et al., T cell development critically depends on prethymic stromal patched expression. J Immunol, 2011. 186(6): p. 3383-91.
  3. de la Roche, M., et al., Hedgehog signaling controls T cell killing at the immunological synapse. Science, 2013. 342(6163): p. 1247-50.

Reviewer 2 Report

The paper is clearly and concisely written and contributes to the field by identifying sub-population of Keratin5+ cells as founding population for skin regeneration. When hedgehog signalling is deregulated in these cells and cancerogenesis is promoted by phorbol esters these cells form BCC lesions. Intriguingly, while cancerogenesis is not promoted, the mutant cells became non detectable and are possibly removed from the skin by physiologically relevant actions of WT cells that surround them. This data are interesting as they postulate that multiple K5+ cells close-by are needed for BCC initiation and since K5+ cells accumulates through age, formation of BCC from these cells becomes more likely, but needs further support from external stimuli. Together, the data nicely match.

There are also only minor spelling errors through the article, e.g.

line 213 supersession - supression.

Author Response

Answer: We reviewed the manuscript and corrected spelling errors throughout the manuscript. The changes were highlighted by grey mark ups.

Round 2

Reviewer 1 Report

I am now convinced of the validity of the conclusions by the additional explanations and additional data presented.